# Breast Cancer Patients: Who Would Benefit from Neoadjuvant Chemotherapies?

**Liqin Yao [1], Gang Jia [2], Lingeng Lu [3,4,5]** and **Wenxue Ma [6,*]**

1. Department of Breast and Thyroid Surgery, The First Affiliated Hospital, School of Medicine, Huzhou University, Huzhou 313000, China; ningmeng2914@sina.com
2. Department of Medical Oncology, Henan Provincial People's Hospital, People's Hospital of Zhengzhou University, Zhengzhou 450003, China; 230159601@seu.edu.cn
3. Department of Chronic Disease Epidemiology, School of Medicine, Yale School of Public Health, New Haven, CT 06520, USA; lingeng.lu@yale.edu
4. Yale Cancer Center, Yale University, 60 College Street, New Haven, CT 06520, USA
5. Center for Biomedical Data Science, Yale University, 60 College Street, New Haven, CT 06520, USA
6. Sanford Stem Cell Clinical Center, Moores Cancer Center, Department of Medicine, University of California San Diego, La Jolla, CA 92093, USA
* Correspondence: wma@health.ucsd.edu; Tel.: +1-858-246-1477

**Abstract:** Neoadjuvant chemotherapy (NACT) was developed with the aims of shrinking tumors or stopping cancer cells from spreading before surgery. Unfortunately, not all breast cancer patients will benefit from NACT, and thus, patients must weigh the risks and benefits of treatment prior to the initiation of therapy. Currently, the data for predicting the efficacy of NACT is limited. Molecular testing, such as Oncotype DX, MammaPrint, and Curebest 95GC, have been developed to assist which breast cancer patients will benefit from the treatment. Patients with an increased level of Human Leukocyte Antigen-DR isotype, tumor-infiltrating lymphocytes, Fizzy-related protein homolog, and a decreased level of tumor-associated macrophages appear to benefit most from NACT.

**Keywords:** breast cancer; cytotoxic T lymphocytes (CTLs); Human Leukocyte Antigen-DR (HLA-DR); neoadjuvant chemotherapy (NACT); pathologic complete response (pCR); tumor-infiltrating lymphocytes (TILs)

## 1. Introduction

There were an estimated 2.3 million new cases of breast cancer worldwide in 2020, accounting for 11.7% of the estimated 19.3 million total new cancer cases [1]. Surgery is still the most important approach among the comprehensive treatment strategies. Adjuvant chemotherapy (ACT), with the goal of reducing the recurrence rate, has been administered since 1976, when Bonadonna et al. reported the successful efficacy of the CMF regimen (cyclophosphamide, methotrexate, and fluorouracil) for patients with lymph-node-involvement breast cancers [2]. Patients with infiltrating breast cancer were at the highest risk.

Breast cancer treatment depends on a variety of factors, such as tumor type, tumor size, metastasis status, and patient preferences. However, the optimal treatment of breast cancers has become more precise according to molecular subtypes, which are classified based on hormone receptor (HR) and human epidermal growth factor receptor 2 (HER2) into four primary molecular subtypes, which include HR+/HER2− (Luminal A), HR+/HER2+ (Luminal B), HR−/HER2− (i.e., triple-negative breast cancer, TNBC), and HER2+. In general, patients with TNBC are more likely to recur than other subtypes, and do not respond well to endocrine-based therapies. Survival rate depends on many different factors, such as stage, type of cancer, grade of the cancer cells, receptor status, and general health. The overall 5-year relative survival rate for breast cancer is 90%, and the 10-year

breast cancer relative survival rate is 84%. Patients with early-stage breast cancer (ESBC) or localized invasive breast cancer have a much higher 5-year survival rate, with 99% for estrogen receptor positive (ER+), 94% for HER2+, and 85% for TNBC [3]. TNBC is more common in women who are younger than 40 years old, have BRCA1 (breast cancer gene 1) mutation, and African Americans. [4]. It has been demonstrated that ACT reduces the recurrence risk and mortality in breast cancer patients [5], and improves overall survival (OS) for the patients with ESBC [6,7]. However, some ESBC patients with ER+ and HER2− may have received unnecessary ACT, and those with a low risk of cancer recurrence may avoid ACT [8]. Thus, accurate prediction of the recurrence risk and response to ACT in patients with ESBC is crucial in order to optimize ACT [9].

Neoadjuvant chemotherapy (NACT) is administered prior to surgery with the goal to shrink tumors and prevent metastasis. NACT has also demonstrated efficacy in downstaging primary tumors and allowing for less morbid surgery instead of mastectomy [10,11]. The chemotherapeutic drugs used for NACT are often the same as for ACT, which, similarly, can cause long-term side effects, such as potentially developing leukemia (rare), cardiomyopathy, osteopenia/osteoporosis, and infertility, as well as short-term side effects, such as cognitive side effects, neuropathy, infection, mucositis, nausea/vomiting, fatigue, and alopecia. Although NACT increases rates of breast conservation, the related serious cost is a higher locoregional recurrence rate [10,12,13]. In fact, only certain types of breast cancer respond especially well to NACT. A study reported that there was no difference in rates of recurrence or OS in breast cancer patients who received NACT compared with those who had ACT based on anthracycline regimens [10].

The aim of this article is to review and summarize the latest research results and address which breast cancer patients would benefit from NACT. Based on current evidence, some breast cancer patients may have similar survival rates without administration of NACT. Consequently, if unnecessary NACT is eliminated, the side effects of NACT can be minimized, while also minimizing the financial burden on governments and patients.

## 2. Adjuvant Chemotherapy (ACT)

Although ACT is beyond the scope of this article, since it is the foundation of NACT, here we provide a quick overview on ACT.

The administration of ACT to patients with non-metastatic, invasive breast cancer is intended to reduce the risk of distant recurrence. Genetic mutations, e.g., BRCA1 and BRCA2, predict the risk of hereditary breast and ovarian cancers, and different genetic signatures have also been developed for precision medicine, helping clinicians and cancer patients to choose the most appropriate therapeutic regimens. Oncotype DX (Genomic Health, Redwood, CA, USA), a gene expression profile consisting of 16 cancer-related genes (*AURKA, BAG1, BCL2, BIRC5, CCNB1, CD68, CTSL2, ERBB2, ESR1, GRB7, GSTM1, MKI67, MMP11, MYBL2, PGR, SCUBE2*), has been officially recommended by the National Comprehensive Cancer Network (NCCN), and widely used to calculate the recurrence score (RS) for ER+ breast cancer patients on a scale of 0–100, with ratings of low (0–10), intermediate (11–25), and high risk (>26) [14]. The MammaPrint (70-gene signature, Agendia Precision Oncology, Amsterdam, The Netherlands) has been approved by the United States Food and Drug Administration (FDA) to calculate RS for both ER+ and ER- breast cancer patients.

In addition, a 95-gene signature assay (Curebest 95GC Breast, Sysmex Corporation, Kobe, Japan), which stratifies patients into high (95GC-H) and low (95GC-L) groups, is used to predict recurrence risk in ESBC patients with ER+, HER2−, and lymph node negative (N0). Curebest 95GC assay has helped reduce the unnecessary administration of ACT [15]. A previous study has also demonstrated that the patients with ER+, HER2−, and N0 have less aggressiveness, and 85% did not experience recurrence [14]. The Trial Assigning Individualized Options for Treatment (TAILORx) was evaluated in a large phase III clinical trial, which compared the combination of ACT and adjuvant endocrine therapy (AET) with AET among ESBC patients with an intermediate RS (11–25). The results found that most cases with ESBC do not benefit from ACT. On this point, Gomez HL et al. reported that up

to 70% of HR (+) and N0 ESBC patients with RS $\leq$ 25 may not need ACT [16]. Since the inconsistencies in risk prediction exist among the genome tests, the European Commission Initiative on Breast Cancer (ECIBC) Guidelines Development Group suggests the use of Oncotype DX for N0 breast cancer patients, and the use of MammaPrint for women at high clinical risk [9].

### 3. Neoadjuvant Chemotherapy (NACT)

The application of NACT increases the opportunity to get more conservative surgery after downstaging the primary tumors, consequently improving the quality of life for women. According to NCCN, patients on NACT may include: (1) locally advanced breast cancer (Stage III, T3 or T4), no matter what the subtypes are; (2) ESBC (Stage I or II) are the most appropriate candidates; (3) patients with a limited number of clinical lymph nodes (+) can undergo sentinel lymph node biopsy, thereby avoiding axillary lymph node dissection and reducing complications such as lymphedema; (4) patients have temporary surgical contraindications such as pregnancy or anticoagulation treatment.

Breast cancer patients with a subtype of ER+, HER2+ are most likely to have no limited response or progression during NACT when compared to the patients with ER−, HER2− subtype (50% vs. 0%, $p = 0.01$) [17].

Although the molecular signature tests were developed for ACT b, their use in NACT has also been explored in the past years, although data to date is limited.

Yardley D et al. enrolled 168 patients with locally advanced HER2− breast cancer (median age 52 years; 45% TNBC) for a NACT study. Oncotype DX was conducted on core needle biopsy of chemotherapy-naive tumor specimens for guiding NACT. Eventually, 161 (96%) patients underwent definitive surgery after NACT. The results showed that the recurrence score (RS) was highly correlated with the achievement of pathologic complete response (pCR, $p = 0.002$), which frequently results in an improved survival [18]. Pease AM et al. identified and included 989 patients (median age, 54.6 years) with ER+, HER2− ESBC, and Oncotype DX assays were used for the choice of NACT. Among the patients, 227 (23.0%) were low RS, 450 (45.5%) were intermediate RS, and 312 (31.5%) were high RS. Only 42 (4.3%) patients achieved pCR. A significant correlation was found between pCR and high RS. This result suggests that Oncotype DX assays could help clinicians to identify the ESBC patients who are the most suitable for NACT [19]. Park KU et al. enrolled 394 primary surgical breast cancer patients who were tested with Oncotype DX assay, including 243 low RS (<18), 125 intermediate RS (18–30) and 26 high RS (>30). The authors constructed a RS predictive model by using Oncotype DX assay results combined with the clinicopathological features (i.e., age, tumor size, histology, ER, PR, and Ki67) to identify high-risk patients (RS > 30). Patients were then assessed for response to NACT. The results showed that 56 patients with high RS received NACT, with 38 patients who responded, and 18 who did not. Oncotype DX assay combined with the clinicopathological features was found to be useful in predicting partial response (PR) to NACT, which can improve the eligibility of lumpectomy [20]. Yao L et al. reported that breast cancer patients with HER2+ or TNBC benefited from standard NACT cycles [21]. Breast cancer patients with the most common type (i.e., invasive ductal carcinoma) at low risk (RS < 10) did not benefit from receiving NACT except AET [15]. Soliman H et al. generated the RS among 764 ER+, HER2− breast cancer patients with Oncotype DX test results and compared to their ability to predict pCR to NACT. The results showed that 59 patients obtained pCR, and those prognostic scores were able to predict response to NACT [22]. Because of its association with improved outcome, pCR to NACT has been accepted as a substitute for surrogate marker for disease-free survival (DFS) and OS in HER2+, TNBC, or luminal B breast cancer patients [22].

There has been a steady increase in the use of NACT for patients with breast cancer, with the highest administration of NACT for patients with HER2+ and TNBC, administered approximately twice as often compared to patients with HR+/HER2− disease [23]. A summary analysis of clinical trials recently concluded that NACT is as effective as the

same therapy as ACT in terms of survival and distant recurrence [13]. NACT has been widely accepted in clinical trials among the patients with HER2+, TNBC, or luminal B due to its association with improved outcome and pCR. The best candidates for NACT are the patients with ER− and/or HER2+, whose pCR rates can approach 65% and predict long-term survival. Patients with ER+, HER2− locally advanced breast cancer are unlikely to achieve pCR from currently available chemotherapy. Consequently, NACT is the best option for ESBC patients, but it remains controversial in ER+/HER2− patients [24].

In summary, Oncotype DX is the most widely used one, and its prognostic value shows a significant benefit for the patients at high risk, and almost no benefit for the low-risk patients. These findings have also been confirmed in the patients with a low-risk cohort [25]. More importantly, the prognostic value of Oncotype DX has also been shown in the breast cancer patients with lymph nodes (+) or locally advanced disease who received NACT [25]. For this reason, these genomic tests are important tools for oncologists, who should be familiar with and should feel comfortable ordering Oncotype DX for their patients [26].

## 4. Biomarkers for NACT

In addition to the RS estimated by multigene signature tests, which is used to determine the likely probability of breast cancer patients in response to NACT, several new immunological biomarkers have recently emerged that could serve as add-ons to help achieve the same goal.

### 4.1. HLA-DR in T Lymphocytes

Major histocompatibility complex class II (MHC II) is expressed by approximately 30% of TNBC cases [27]. Forero A et al. reported that an aberrant expression of MHC II molecules in TNBC cancer cells may trigger an anticancer immune response, which reduces the recurrence rate, and thus, improves progression-free survival (PFS). Therefore, high MHC II gene expression and increased tumor-infiltrating lymphocytes (TILs) in TNBC patients is associated with favorable prognosis [28]. It was found that MHC II + breast tumors have a higher degree of TILs after NACT, which correlates with improved survival after surgical resection [29]. MHC II is an important biomarker in breast cancer [30]. Human Leukocyte Antigen-DR isotype (HLA-DR) is an MHC II cell surface receptor, which is encoded by human leukocyte antigen (HLA) complex on chromosome 6 region 6p21.31. HLA-DR is an antigen-presenting molecule, typically expressed on professional antigen-presenting cells (APC) and is associated with antigen presentation. However, it is also expressed on activated T lymphocytes in the circumstance of autoimmune diseases, viral infections, and cancer. Thus, HLA-DR might be a useful biomarker for identifying effector T cells and monitoring immune responses during breast cancer treatment.

In addition to an increased level of HLA-DR in infection diseases, Saraiva D et al. investigated the clinical relevance of HLA-DR in breast cancer patients. In the study, the authors collected 48 fresh biopsies, 96 non-matched surgical samples from breast cancer patients, 31 matched peripheral blood samples from the breast cancer patients, and 18 peripheral blood samples from healthy donors as the controls. The results from flow cytometry analysis showed that a high level of HLA-DR expression was only found in the cytotoxic T lymphocytes (CTLs) derived from the N0 breast cancer patients and responders to NACT ($p < 0.01$). Additionally, the authors observed that the expression of HLA-DR in TILs (i.e., CTLs) was highly correlated with that in circulating CTLs ($p = 0.001$). More interestingly, higher levels of HLA-DR (>8.943, a cut-off value) were found in responders to NACT when compared to non-responders and healthy donors ($p < 0.05$). IFN-$\gamma$ levels in breast cancer patients with both N0 and responders to NACT were significantly higher when compared to patients with lymph nodes (+) and non-responders, respectively ($p < 0.01$, $p < 0.05$). Furthermore, IL-10 levels in patients with lymph nodes (+) was significantly higher compared to the N0 patients ($p < 0.05$). Consequently, the authors believe that HLA-DR level in CTLs is a highly sensitive and specific potential predictive marker to NACT response, which can be assessed in peripheral blood to guide therapeutic decisions [31]. Based on their

previous results, the authors carried out further research and enrolled 202 breast cancer patients, which included 61 biopsy specimens, 41 blood samples before receiving NACT, and 100 non-NACT tumor samples. All the patients enrolled in this study were followed for up for 34 months. The authors confirmed that high HLA-DR level (above the cutoff value, ranging 8.94–9.3 in different cohorts) in CTLs is a highly sensitive, specific, and independent biomarker to predict improved response to NACT [32]. HLA-DR is now used as a marker of T cell activation [33,34]. Saraiva DP et al. reported that the CTLs expressing high levels of HLA-DR are mainly located in intraepithelial tumor structure. The CTLs from the breast cancer biopsies between the responders and non-responders to NACT were analyzed by flow cytometry. The results showed that only the NACT responders with N0 have high levels of CTLs expressing HLA-DR [32].

Moreover, Stewart RL et al. studied the immune activation score of the MHC II molecule in 44 primary TNBC patients. The results demonstrated that an immune activation score of MHC II was significantly associated with longer DFS of the TNBC patients ($p$ = 0.01) [35]. In their following study, the authors included an additional 10 TNBC patients to quantify HLA-DR expression with NanoString Digital Spatial Profiling (DSP) and confirmed that HLA-DR expression in tumor interstitial lymphocytes was associated with longer DFS compared to the patients with low HLA-DR expression. [36]. One possible way to improve the selection of patients who would benefit from NACT would be to perform profiling on both immune cells and the primary breast tumor (Figure 1).

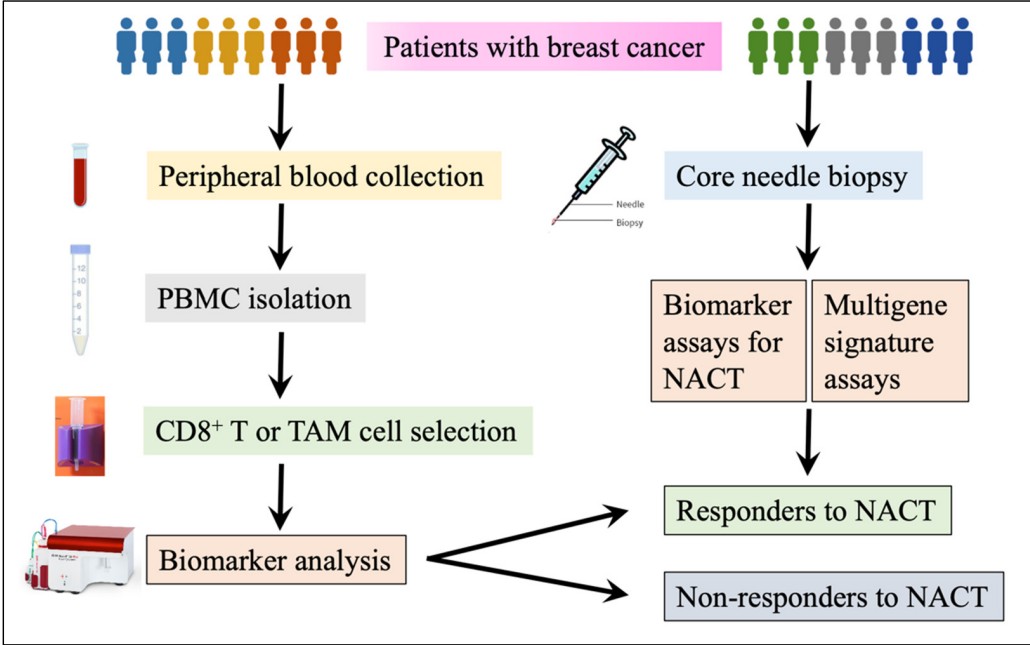

**Figure 1.** A flow chart for multigene and biomarker assays to predict NACT response. Peripheral blood or biopsy samples can be obtained from a blood drawn or core needle biopsy. CD8+ T or TAM cells are isolated from PBMCs and are harvested after a density gradient centrifugation with Ficoll-Paque. Biomarker assays are completed with the aid of a flow cytometer, quantitative polymerase chain reaction (qPCR, also known as RT-PCR), *immunohistochemistry* (IHC), or Western Blot as needed. For multigene signature assay, it is much easier as a commercial service: obtain a special specimen transportation kit, then complete a requisition form, submit samples, and finally access and get the test results.

### 4.2. Tumor-Infiltrating Lymphocytes (TILs)

TILs are activated T lymphocytes, especially CD8+ T cells (i.e., CTLs) that infiltrate tumor tissue, and potentially eliminate cancer cells. There is increasing evidence that TILs are a positive prognostic biomarker in TNBC. Specifically, an increased presence of TILs in the tumor microenvironment leads to a better prognosis, and a better survival [29]. However,

cancer cells can evade immune surveillance by crippling CTL functionality via the production of several immune suppressive cytokines. Cancer cells express the inhibitory molecule, programmed death-ligand 1 (PD-L1), which binds to the inhibitory receptor PD-1 on CTLs, and thus, inhibits CTLs' activity. Therefore, a negative regulatory pathway drives CTLs into an exhaustive state, which has been shown in association with poor prognosis [37]. Other inhibitory immune checkpoint (e.g., CTLA-4) and immunosuppressing molecules, such as IL-10, TGF-β, or IDO (indoleamine 2,3-dioxygenase), could also negatively impact CTL's activity [38] and infiltration into tumors [39]. Iwamoto K et al. reported that TILs have a predictive value for prognosis and response to chemotherapy [40]. Stewart RL et al. also showed that the expression of MHC II mRNA in TNBC samples is correlated with the presence of TIL genes, and that a high MHC-II immune activation score predicted a better prognosis of TNBC [35] (Figure 2).

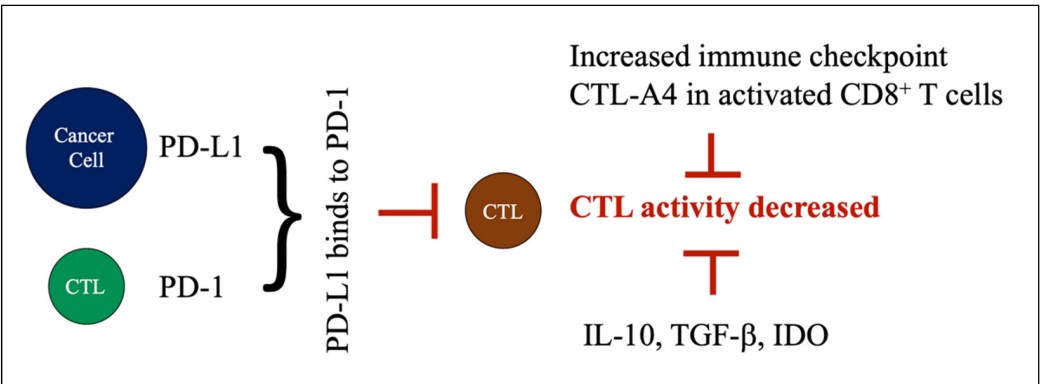

**Figure 2.** Factors decreasing CTL cell activity. Cancer cells express inhibitory molecule PD-L1, which binds to PD-1 on CTLs. This binding induces a negative regulatory pathway that limits CTL activity. Other inhibitory immune checkpoints, such as CTL-A4, and immunosuppressing molecules (i.e., IL-10, TGF-β, IDO) could also negatively impact CTL activity.

### 4.3. Tumor-Associated Macrophages (TAMs)

Tumor-associated macrophages (TAMs) are *macrophages* that participate in the formation of the *tumor* microenvironment by producing cytokines, chemokines, and growth factors that can be immunosuppressive to CTLs, leading to tumor progression and metastasis [41,42]. Patients with a high abundance of infiltrated M2 macrophage have a shorter OS compared to those with a low abundance [39]. However, the role of TAMs in NACT is unknown.

Ye JH et al. included 91 primary TNBC patients (23–65 years) treated with NACT in their study and investigated if TAMs are related to the NACT response. The patients had either locally advanced or unresectable breast preservation. During the follow up of 45.8 months on average, 16 of the 65 non-pCR cases relapsed, including local recurrence and distant metastasis. The results showed that the OS and recurrence-free *survival* (RFS) rates were significantly lower among the patients with high infiltration of TAMs (the median of infiltrated TAMs as a cut-off value) than those with low infiltration ($p = 0.023$ and $p = 0.013$, respectively). Survival data showed that patients with high infiltration of CD163+ macrophages and non-pCR had poor OS and RFS [43]. Therefore, TAMs may be used as a promising prognostic candidate biomarker to predict NACT response. Ni C et al. conducted a meta-analysis of 13 studies including 5116 patients and demonstrated that high CD163+ TAMs were associated with poor OS ($p = 0.003$). This result further confirmed the clinical significance of TAMs in breast cancer [44]. High CD163+ TAMs could be used as a promising prognostic biomarker in non-metastatic breast cancer to predict poor NACT response.

### 4.4. Fizzy-Related Protein Homolog (FZR1)

Fizzy-related protein homolog (FZR1), also known as cell division cycle 20 related 1 (Cdh1), is an activator of anaphase promoting complex/cyclosome (APC/C), and functions as E3 ubiquitin ligase that drives the cell cycle [45]. APC/C is a multifunctional ubiquitin protein ligase, which targets different substrates, including NEK2A, Cyclin A, Cyclin B, and Securin [46], and a regulator of chromosome segregation (RCS1) [47] for ubiquitylation, and consequently regulates cell division, differentiation, cell death and carcinogenesis, etc. [48]. It is a key regulator of cell mitosis and G1-phase in the cell cycle. In addition, tumor suppressor retinoblastoma protein (pRB) also plays a key role in cell cycle regulation and inhibits proliferation by inhibiting cell transition from G1 to S-phase. Furthermore, pRB and FZR1 are also involved in cell differentiation, dormancy, apoptosis, and maintaining chromosome integrity and metabolism [49]. The I et al. reported that simultaneous deletion of pRB and FZR1 synergistically bypassed cell division arrest in human breast cancer cells [50]. The roles of APC/FZR1 (i.e., APC/Cdh1) and APC/C are summarized in Figure 3.

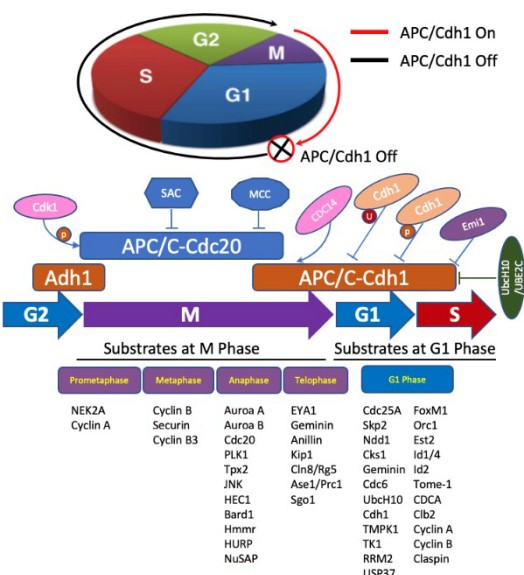

**Figure 3.** Role of APC/FZR1 and APC/C in cell cycle regulation. APC/C regulation of cell cycle progression occurs mainly through Cdc20 or Cdh1 temporal coordination. APC/C-Cdc20 (blue) is activated by Cdk 1 phosphorylation, inhibited by spindle assembly checkpoint (SAC) and mitotic checkpoint complex (MCC), and degrades substrates in early/middle mitosis. APC/C-Cdh1 (brown) degrades substrates during the end of mitosis and G1 phase. When cell commences to anaphase, Cdh1 is dephosphorylated by CDC14 and activates APC/C-Cdh1, which ubiquitylates substrates during anaphases and telophase. APC/C-Cdh1 degrades mitotic cyclins at G1 phase, and then it is inactivated by Emi1, Cdh1 degradation, and phosphorylation by cyclin A/Cdk2 and degradation of E2s during G1/S phase transition.

To investigate the function of FZR1 in in breast cancer patients with NACT, Liu S et al. collected 193 specimens from breast cancer patients who underwent NACT, including 14 cases of Luminal A, 122 cases of Luminal B, 45 cases of HER2+, and 12 cases of Basal 12. IHC staining was conducted to identify the expression of FZR1 on the samples of formalin-fixed paraffin embedded breast cancer tissues. The results showed that the sensitivity and specificity of FZR1 expression (IHC score of 17.5 as a cut-off value) to NACT response was 97.01% (CI: 89.63–99.94%) and 98.39% (CI: 91.34–99.96%), respectively. These data demonstrated that FZR1 is an efficient predictive biomarker for NACT response. The IHC score of FZR1 examination can be used in clinics to evaluate the effect of NACT, which will provide a better therapy for patients. It is demonstrated that FZR1 is pivotal to

chemotherapy-induced apoptosis and cell cycle arrest [51]. Thus, the authors recommend that the FZR1 IHC score be used to predict the efficacy of NACT [52].

## 5. Discussion

Breast cancer is one of the most diagnosed cancer types in women. NACT is widely used for breast cancer treatment to shrink and downstage tumors, as well as stop cancer cell dissemination. Topics on chemotherapy options, evaluation, and treatment after NACT, as well as the clinical prognostic staging system including these prognostic features, such as tumor grade, status of hormone receptors (estrogen receptor and progesterone receptor, i.e., ER and PR), and HER2, can be found from the latest clinical guidance, such as 2021 NCCN, or from the eighth edition of the American Joint Committee on Cancer (AJCC). According to the guanaclines, NACT is appropriate for the patients with locally advanced breast cancer regardless of biologic subtype, who are generally classified at stage III and stage IIB cancers, and T3 disease [53]. In this article, we mainly described the selection of breast cancer patients based on multigene genomic tests and biomarker assays, and then the decision of which patients would benefit from NACT. Some prognostic features, such as tumor grade and receptor status of both hormones and HER2, provide more refined prognostic information before surgery.

According to 2019 ASCO meeting presentations, it was reported that 70% of the breast cancer patients with HR+, HER2−, and N0 can forego chemotherapy when guided by recurrence score (https://ascopost.com/news/60141, accessed on 13 June 2019). For the breast cancer patients receiving NACT, it is a trade-off between side-effects and survival benefit. The ideal choice is complete response (CR) with little side effects. Consequently, a couple of multigene signature assays have been used for the precision medicine of both ACT and NACT.

Oncotype DX is a 21-gene signature test that primarily relates to estrogen signaling, which influences the aggressiveness of breast cancer. MammaPrint uses a 70-gene signature test to calculate RS on both ER+ and ER− breast cancer patients at either low or high risk. The difference between these two genomic tests is the percentage of breast cancer patients classified as a low risk of recurrence. Oncotype DX assay not only measures the recurrence possibility of the breast cancer patients with ER+, N0, but also predicts the degree that the patients will benefit from ACT [54]. Oncotype DX detected a lower percentage of patients with ESBC who are at low risk, whereas MammaPrint detected a higher one [55]. MammaPrint is able to identify more patients with ESBC who might not need chemotherapy. One study shows that the MammaPrint test may eventually be widely used to help make treatment decisions based on the risk of cancer recurrence within 10 years after diagnosis [56]. Another study by Cardoso F et al., who enrolled 1550 patients with ESBC, determined the genomic risk with the MammaPrint test, and compared their clinical risk in selected patients for ACT. Their results demonstrated that the 5-year distant metastasis-free survival rate in the ESBC patients with high clinical risk but low genomic risk who received ACT was only 1.5% lower than that in those patients without ACT [57]. Coincidentally, similar survival rates were also found in the distant metastasis-free survival of patients with ER+, HER2−, and lymph node (+) or N0. Collectively, approximately 46% of patients with ESBC who are at high clinical risk might not require ACT [58]. These genomic tests are commonly used for ACT, but their use in NACT is still limited. Based on the current limited data for NACT, Oncotype DX assay is the most widely used in clinics. The results showed that high-risk patients with breast cancer benefit from NACT, and low-risk patients almost do not.

The selection of breast cancer patients on NACT is critical. Non-effective NACT could increase the risks of delaying surgery and may lead to the development of unresectable disease and metastatic tumor spread. To individualize treatment, reduce unnecessary morbidity, and improve the treatment outcomes, it is very important for surgical oncologists to understand how to introduce and use these genomic test tools into their clinical practice for patients with breast cancer. In addition to the use of multigene assays, it is greatly

needed to identify useful biomarkers for the efficiency of NACT, distinguishing the non-responders from responders. In this article, in addition to the description of application of multigene signature assays, we also summarized the useful biomarkers for breast cancer patients who may benefit from NACT, including HLA-DR expression in CTLs, which is a novel biomarker that can be used in ESBC. TILs have a predictive value for prognosis and response to chemotherapy. CD163[+] TAMs can also serve as a promising prognostic candidate biomarker for predicting response to NACT treatment. FZR1 is an efficient biomarker for NACT effect prediction as well.

No matter how successful and efficacious genomic tests and biomarker assays are, there are still some potential limitations due to tumor heterogeneity. All the samples are taken from biopsies for either multigene tests or biomarker assays for precision medicine. First, the volume (size) of biopsy samples is very limited, especially when using fine needle aspiration, a minimally invasive biopsy. Core needle biopsy is a larger needle that would be more helpful to remove a larger tissue sample. Both fine needle aspiration and core needle biopsy are more common for deep lesions that are not superficially located, to allow for simple surgical excision. In addition, excisional biopsy can obtain more suspicious cancer tissues. All these common biopsy techniques may need an aid with image guidance, including ultrasound, fluoroscopy, computed tomography (CT) scan, X-ray, or magnetic resonance imaging (MRI) scan, when a tumor is not palpable, or the location is deeper. The success of the biopsy depends on the skill of the clinicians. These tests, including multigene signatures and biomarker assays, cannot be smoothly completed if the biopsied tumor tissues are not adequate.

This article itself may have some limitations. The investigations of precision NACT on breast cancer patients are limited. Among these limited studies, some might have selection bias in the enrollment of patients or surgical delay for patients with NACT. In addition, the research works on multigene signatures and the above-discussed biomarker assays in breast cancer patients receiving NACT are still relatively inadequate. It is hoped that more clinicians and relevant researchers will keep increasing their efforts in this field, and the results can be further used for guiding clinical practice.

## 6. Conclusions

Some of the ESBC patients with ER+ and HER2− may have received unnecessary NACT, which might not add an additional benefit beyond surgery. The patients at low risk of cancer recurrence may directly undergo surgery without NACT.

Among the genomic signature tests, Oncotype DX is the most widely used one. Its prognostic value shows a significant benefit for breast cancer patients at high risk of RS, and almost no benefit for the patients at low risk of RS. Oncotype DX assay can be used not only in patients with N0, but also in the patients with lymph nodes (+) or locally advanced disease to predict if the patients could benefit from NACT.

Biomarker assays may help to precisely identify who will benefit from NACT and who will not. Patients with either high levels of HLA-DR or TILs, or high activation score of MHC-II, or low level of CD163[+] TAMs, as well as high expression level of FZR1, had better response to NACT in breast cancer. Moreover, the association of these immune response-related biomarkers with NACT response suggests that the immune system may be involved in patients' response to NACT, and that the synergistic effects between NACT and immunotherapy may exist.

**Author Contributions:** Conceptualization, W.M.; writing—original draft preparation, L.Y., G.J. and W.M.; writing—review and editing, L.L. and W.M. All authors have read and agreed to the published version of the manuscript.

**Funding:** This research received no external funding.

**Acknowledgments:** The authors would like to thank Benjamin Heyman at Moores Cancer Center, University of California San Diego Health for his great help with critical reading and editing. This paper reflects the opinions in the related publications and current literature accessed by the authors; no formal search strategy has been defined.

**Conflicts of Interest:** The authors declare no conflict of interest.

## Abbreviations

ACT, adjuvant chemotherapy; AET, adjuvant endocrine therapy; BRCA, breast cancer gene; DFS, disease-free survival; ESBC, early-stage breast cancer; ER, estrogen receptor; ESBC, early-stage breast cancer; FZR1, fizzy-related protein homolog; HER2, human epidermal growth factor receptor 2; HR, hormone receptor; IHC, immunohistochemistry; NACT, neoadjuvant chemotherapy; pCR, pathologic compete response; PFS, progression-free survival; N0, lymph node negative; NCCN, National Comprehensive Cancer Network; OS, overall survival; PR, progesterone receptor; PR, partial response; RFS, recurrence-free survival; RS, recurrence score; TAILORx, Trial Assigning Individualized Options for Treatment; TAM, tumor-associated macrophage; TNBC, triple-negative breast cancer.

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
