# Peer review of "Breast Cancer Patients: Who Would Benefit from Neoadjuvant Chemotherapies?"

_curroncol, doi:10.3390/curroncol29070389_

Round 1
Reviewer 1 Report
The work tries to summarise based on few studies and tries to draw conclusion that can be beneficial for the clinicians. Biomarkers for NACT is discussed quite well. However, major focus is given to TIL and HLA-DR. Many new biomarkers have evolved and use of FZRI is also increasing. Authors can detail about FZRI with appropriate scheme and diagrams.
Few abbreviations are not mentioned in the abbreviation section and directly used in the text without expanding anywhere in the text. eg. N0. Such things needs to avoided or appropriately mentioned in the text or abbreviation section.
Author Response
The work tries to summarize based on few studies and tries to draw conclusion that can be beneficial for the clinicians. Biomarkers for NACT is discussed quite well. However, major focus is given to TIL and HLA-DR. Many new biomarkers have evolved and use of FZRI is also increasing. Authors can detail about FZRI with appropriate scheme and diagrams.
Authors: We are grateful to reviewer 1 for his/her constructive comments and suggestions on the biomarker of FZR1. For the minor problems that may exist in language and style, we have asked a native English speaker, Dr. Benjamin Heyman (MD, PhD) who is an Oncologist/Hematologist at The Rebeca and John Moores Cancer Center, University of California San Diego to help with editing the entire manuscript.
To be sure, the articles related to FZR1 in chemotherapy or NACT for breast cancer patients are still very limited after double-searching. But reviewer 2's suggestion is an excellent idea. To help the potential readers to better understand FZR1, a schematic diagram was added in the revised version to illustrate the role of FZR1 in cell cycle.
Few abbreviations are not mentioned in the abbreviation section and directly used in the text without expanding anywhere in the text. eg. N0. Such things needs to avoided or appropriately mentioned in the text or abbreviation section.
Authors: We appreciate the reviewer 1’s comments on the abbreviations. We have edited and added the full names of the terms and their corresponding abbreviations not only in the revised text, but also in the abbreviation list based on alphabetic order.

Reviewer 2 Report
Neoadjuvant chemotherapy shrinks tumors and stops cancer cells from spreading before surgery, but sometimes there are the side effects and not all patients benefit from it. Molecular signatures of Oncotype DX, MammaPrint, and Curebest 95GC have been further developed to determine which patients would benefit from this treatment. The patients with an increased level of Human Leukocyte Antigen-DR isotype, tumor-infiltrating lymphocytes, Fizzy-related protein homolog, and a decreased level of tumor-associated macrophages would benefit from neoadjuvant chemotherapy. Accordingly to authors, patients who don't meet these criteria may skip neoadjuvant chemotherapy so that they may avoid the unnecessary side effects, delaying surgery, and reduce the medical cost burden.
This manuscript is clear, comprehensive, well-written and structure. Figures are very schematic and clarifiers to understand the text.
The main subject of the study is well-defined and conclusions are consistent with the arguments presented in Discussion.
It has current and relevant references (most of them between 2017 and 2022) and there are not excessive number of self citations (only 4/52).
In general, this paper is scientifically sound. The issue is relevant for the field, authors made a good job evaluating this therapy around breast cancer.
I consider that this work fit the Current Oncology journal scope.
Author Response
We appreciate the viewer 2's very positive and confirmed comments on the article.
For the minor issues that may have in language and style, we have asked Dr. Benjamin Heyman (MD, PhD) who is not only a native English speaker, but also an Oncologist/Hematologist at The Rebeca and John Moores Cancer Center, University of California San Diego to help with editing this manuscript. We would like to thank Dr. Heyman for taking the time out of his busy daily clinical schedule to revise this article
